# COVID-19 Vaccination Strategies and Their Adaptation to the Emergence of SARS-CoV-2 Variants

**DOI:** 10.3390/vaccines10060905

**Published:** 2022-06-06

**Authors:** Paola Stefanelli, Giovanni Rezza

**Affiliations:** 1Department of Infectious Diseases, Istituto Superiore di Sanità, 00161 Rome, Italy; 2Directorate of Health Prevention, Ministry of Health, 00144 Rome, Italy; g.rezza@sanita.it

**Keywords:** SARS-CoV-2, COVID-19, vaccines, vaccination strategies, variants

## Abstract

About one year after the identification of the first cases of pneumonia due to a novel coronavirus in Wuhan, several vaccines against SARS-CoV-2/COVID-19 started to be approved for emergency use or authorized for early or limited use. The rapid development of effective vaccines based on different technological platforms represents an unprecedented success for vaccinology, providing a unique opportunity for a successful public health intervention. However, it is widely known that only a limited number of vaccine doses are usually available at the beginning of vaccination campaigns against an emerging virus; in this phase, protecting health care workers and reducing mortality rates is the priority. When a larger number of vaccines become available, the identification of the drivers of virus circulation coupled with the use of transmission blocking vaccines are key to achieve epidemic control through population immunity. However, as we learned during the vaccination campaigns against the pandemic coronavirus, several factors may hamper this process. Thus, flexible plans are required to obtain the best sustainable result with available tools, modulating vaccination strategies in accordance with improved scientific knowledge, and taking into account the duration of protective immune response, virus evolution, and changing epidemic dynamics.

## 1. Tackling the COVID-19 Epidemic: The Race of Vaccines

On 31 December 2019, cases of pneumonia of unknown origin were reported in the city of Wuhan, China [1]. The number of cases increased rapidly, as well as outside China, and on 11 March, the World Health Organization declared COVID-19 (coronavirus disease 2019, caused by SARS-CoV-2) a pandemic (https://www.who.int/director-general/speeches/detail/who-director-general-s-opening-remarks-at-the-media-briefing-on-covid-19---20-march-2020).

Different from influenza, there were no existing vaccines or production processes for coronavirus vaccines at the beginning of the pandemic. However, previous studies on other emerging coronaviruses, such as SARS-CoV-1 and MERS-CoV, had shown that the S protein on the surface of the virus might be an ideal target; in fact, since the spike protein interacts with the ACE2 receptor, antibodies targeting this protein may interfere with this binding and neutralize the virus [2].

Thanks to the mobilization of human and financial resources, favored by the commitment of governments and private companies, during the first year of the pandemic, there was an acceleration in the development of COVID vaccines, leading to an unexpected success that has been correctly defined as a “triumph” for vaccinology [3].

Several platforms were utilized to develop vaccine candidates: genetic (mRNA and synthetic DNA-based), viral vector, protein-based, whole virus attenuated or inactivated vaccines, and virus-like particles [4]. By 3 January 2022, of 33 vaccine candidates in phase 1, 20 in phase 1/2, 17 in phase 2, 11 in phase 2/3, and 38 in phase 3, 19 vaccines were authorized for early or limited use, and 12 were approved for full use, while 10 candidates were abandoned after trials. Moreover, at least 75 pre-clinical vaccines were under animal experimentation (https://www.nytimes.com/interactive/2020/science/coronavirus-vaccine-tracker.html, accessed on 18 March 2022). Of the authorized vaccines, mRNA, adenovirus vectored vaccines, and protein-based vaccines have been so far commercialized in the UK, US, and EU, whereas inactivated vaccines are widely used in other areas of the world.

Since several vaccines are currently available, the implementation of optimal strategies is crucial in maximizing the benefits of their use. It is also important to learn from the experience of countries that firstly immunized a large proportion of the population, with extremely positive results in terms of population impact and estimated vaccine effectiveness against symptomatic disease, hospitalization, and death [5,6]. In this conceptual, non-systematic review, we discuss vaccination strategies against COVID-19, focusing the attention on bottlenecks and hurdles that may jeopardize the achievement of epidemic control, with particular regard to the role of virus variants.

## 2. Designing a Vaccination Campaign

The goals of a vaccination campaign against SARS-CoV-2/COVID-19 consist of: (i) reducing SARS-CoV-2 transmission and/or morbidity and the mortality caused by COVID-19 disease, (ii) reducing the pressure on the healthcare system and maintaining its efficiency; and (iii) minimizing social and economic disruption and favoring the re-opening of the society (https://www.ecdc.europa.eu/sites/default/files/documents/Objectives-of-vaccination-strategies-against-COVID-19.pdf, accessed on 23 April 2021).

Different strategies may be used to achieve these goals. In general terms, vaccination strategies may focus on direct protection (i.e., vaccinating the elderly and people with comorbidity) or indirect protection (vaccinating those in contact with high-risk individuals, age groups acting as epidemic amplifiers, or the community at large). Whether available vaccines are able only to protect at-risk individuals from severe disease or if they may even prevent infection and/or reduce contagiousness is key to deciding the best approach [7].

A multi-phased approach in vaccination programs is frequently required. In fact, there is usually a short period of time when limited doses of a vaccine are available only for highly targeted population groups; in this phase, priorities should be set. After that phase, a larger number of doses becomes available, requiring a broader administration network (https://www.cdc.gov/vaccines/acip/meetings/downloads/slides-2020-08/COVID-08-Dooling.pdf, accessed on 26 August 2020).

In detail, vaccination strategies against severe emerging infections may be based on non-mutually exclusive conceptual approaches: (i) targeting priority groups at risk of severe disease and mortality, such as older adults and/or individuals with underlying conditions; (ii) targeting priority groups at higher risk of exposure and/or essential services, such as healthcare workers (HCWs); (iii) aiming at an efficient reduction in disease transmission at the population level (i.e., population groups representing the main drivers of transmission); (iv) targeting high incidence areas/regions and/or outbreak settings; and (v) universal vaccination (https://www.cdc.gov/vaccines/covid-19/phased-implementation.html (accessed on 15 January 2021). In any case, an adaptive approach based on the evidence emerging during the course of the epidemic should always be considered.

Most countries have implemented COVID-19 vaccination campaigns starting with the elderly and other individuals at risk (because they are affected by comorbidity) using a sequential approach based on different phases with specific aims. (i) Phase 1.0: to reduce mortality and severe morbidity and maintain the healthcare system capacity (the vaccination of HCWs and care-givers is included in this phase); (ii) phase 2.0: to reduce or eliminate virus circulation as much as possible; and (iii) phase 3.0: to boost the immune response and reduce the risk of emerging variants possibly through multivalent vaccines.

### 2.1. Delaying the Second Dose of Vaccine

In accordance with the results of vaccine trials, a two-dose primary cycle was adopted for most vaccines. However, alternative vaccination schemes were implemented in some countries in order to accelerate the vaccination campaign and to reduce mortality in the short-term. In particular, administering the first of two vaccine doses to as many people as possible to provide a stronger population effect in the short term was recommended in the UK to try to mitigate the rapid spread of the highly transmissible SARS-CoV-2 Alpha variant [8]. In this regard, studies conducted in Israel showed that a single dose of the mRNA COVID-19 vaccine was immunogenic in the vast majority of vaccinated individuals [9], and that a substantial reduction in both SARS-CoV-2 infection and symptomatic COVID-19 rates may be observed following the first vaccine dose [10], with risk/rate reductions in SARS-CoV-2 infections ranging from 51 to 75% [10,11]. Furthermore, the mass roll-out of the first dose of Comirnaty and Vaxzevria in Scotland was associated with a reduction in the risk of hospital admission of 91 and 88%, respectively [12]. However, some critical remarks came from the results of mathematical models suggesting that the single/delayed dose strategy may decrease infections in the short term but is not necessarily successful in the long term [13]. Finally, the emergence of variants with partial immune escape indicated that anticipation of the administration of the second dose is required to increase protection against infection at the individual and community level (see later).

### 2.2. Increasing Vaccines Availability and Confidence

In order to plan campaigns able to protect as many people as possible, the vaccine manufacturing capacity needs to be scaled up, from the production of millions to a billion doses of the vaccine, and potential constraints such as costs, distribution systems, cold chain requirements, and delivery should be alleviated [14].

The willingness to be vaccinated is also a key factor to achieve high vaccination coverage. Surveys conducted at the beginning of the vaccination campaign found that 74% of the Europeans but only 49% of the Americans surveyed declared that they would be willing to be vaccinated [15,16]. However, 55% of the Europeans said they were concerned about vaccines’ side effects. Actually, safety concerns related to specific vaccines led governments to issue warnings and precautions, affecting decisions on recommendations targeting specific age groups. Finally, vaccine acceptance was found to be lower in specific population groups [17]. These findings raised some concern, since it is critical to ensure a high enough vaccination coverage to achieve herd immunity. To this end, planning education programs is important in order to eliminate prejudice and misconceptions about vaccines. However, vaccination rates higher than 80% were reached in several European countries at the beginning of the year 2022 (https://vaccinetracker.ecdc.europa.eu/public/extensions/covid-19/vaccine-tracker.Html#uptake-tab, accessed on 18 March 2022). Yet, beyond the youngest age groups, a relatively small proportion of the population at risk may refuse to be vaccinated. Whether mandatory vaccination should be considered to contrast vaccine hesitancy, at least for special population groups, remains a controversial issue [18].

## 3. Epidemic Control and the Herd Immunity Threshold

Obtaining a high level of epidemic control or even herd immunity is an important public health goal of mass vaccination programs. To this end, a large proportion of the population needs to be vaccinated; moreover, the identification and immunization of epidemic drivers (i.e., age groups that sustain SARS-CoV-2 transmission) through the use of transmission-blocking vaccines may accelerate epidemic control.

The herd immunity threshold depends on the R_0_ of the disease and was estimated to range between 50% and 67% for SARS-CoV-2 at the beginning of the vaccination campaign, when the original Wuhan strain was still circulating [19]. This threshold may become even higher in the case of the circulation of new variants with higher transmission potential (and higher R_0_). However, even high vaccination coverage may not ensure the complete control of the epidemic; that depends on a series of factors that represent possible hurdles to control and/or elimination programs: (i) the limited capacity of available vaccines to induce sterilizing immunity; (ii) the short duration of protection; (iii) the emergence, introduction, and circulation of immune escape variants; (iv) the lack of vaccines or their limited use among children and adolescents; and (v) poor access to vaccines in poor-resource countries [20].

Although it has been suggested that mass vaccination with transmission-blocking vaccines among adults aged 20 to 49 is likely to bring the epidemic under control [21], mathematical models indicate that vaccination campaigns that prioritize those at risk of death are the ones that avert most deaths, even in the case of vaccines extremely effective in blocking transmission [22]. Thus, target group prioritization followed by mass vaccination appears to be the wiser strategy to be adopted. In this regard, it is likely that epidemic control (or at least a return to normality) without non-pharmaceutical interventions might be, to some extent, possible only with a highly effective variant-matched vaccine, very high vaccination coverage, and extending vaccination to children [23].

### 3.1. Sterilizing Immunity and Protection from Severe Disease

A key question closely connected with herd immunity is whether currently available vaccines are capable of blocking virus transmission through the induction of sterilizing immunity or if they only confer protection against severe disease. Preventing infection through vaccination in a relatively high proportion of cases is considered a possible goal of immunization programs. A nationwide study conducted in Israel found vaccine effectiveness against SARS-CoV-2 infection at 7 days or longer after the second dose of mRNA vaccines to be 95.3%, ranging from 91.5% to 97.0% against asymptomatic infection and COVID-19, respectively [24]. Data from the UK showed relatively high effectiveness against infection and against transmission for both mRNA and viral vectored vaccines, showing up to a 50% reduction in secondary cases in the household of a symptomatic index case [6]. Moreover, other studies found the viral load to be substantially reduced for breakthrough infections occurring 12 to 37 days after the first dose of an mRNA vaccine, suggesting a lower infectiousness that may contribute to the vaccines’ effects on the spread of SARS-CoV-2 [25]. However, these findings are not confirmed when variants with partial immune escape are considered. Thus, population immunity could be hardly achieved, even with extremely high vaccination coverages [26]. In a few words, currently available vaccines may provide sterilizing immunity against vaccine-matched virus strains. However, when variants with partial immune escape emerge, these vaccines mostly confer protection against severe disease.

### 3.2. Waning Immunity

Animal models showed that experimental infection with SARS-CoV-2 protects against reinfection in rhesus macaques [27,28]. However, the occurrence of sporadic cases of reinfection suggested that such protection can be of short duration, at least in human beings [29,30,31]. This finding was initially explained by a rapid decay of neutralizing antibody levels after infection [32,33,34], which appeared to be similar to the decline observed with other human coronaviruses, such as NL63, 229E, OC43, and HKU1 [35]. However, other studies showed that serum-neutralizing antibodies induced by an mRNA vaccine (i.e., mRNA-1273) may continue to persist for at least 6 months after the second dose, albeit at lower levels compared with the time of the peak of the immune response [36,37,38]. Furthermore, a study conducted in Israel found an increased risk of infection for those vaccinated in January compared with those who were vaccinated in April 2021, confirming waning immunity as a factor of time from vaccination [39]. However, though the waning of neutralizing antibodies is to some extent worrying, robust T cell response and B cell memory may still protect against severe disease over the longer term [40,41]. In any case, booster doses have been recommended to maintain a high level of protection even against emerging variants (see below).

In conclusion, there is mounting evidence that both natural infection and vaccination may induce a durable immune response against severe disease but not against infection. How long vaccine-induced protection may last depends on a series of factors that need to be examined in detail.

### 3.3. The Emergence of Immune Escape Variants

Since RNA viruses are likely to mutate during the course of epidemics, the possible impact of virus drifts on vaccine efficacy should be always considered. However, only sometimes does viral evolution erode immunity, and different viruses behave in different ways. For example, in the case of measles, the immune response is equally potent against all strains and does not generate selection, whereas partial immunity to influenza A virus generates strong fitness differences among strains, leading to rapid strain turnover due to continual immune selection [42]. This may explain why the measles vaccine maintains its efficacy over time, while the influenza vaccine needs to be adapted and updated almost every year.

With regard to SARS-CoV-2, some variants that emerged during the first epidemic wave, such as the one carrying the D614G mutation in the immunogenic spike glycoprotein, appeared to be associated with enhanced transmission but not with reduced vaccine efficacy [43]. Other variants that have been later identified, such as the UK strain VOC 202012/01 (so-called Alpha variant), were found to be more transmissible than previously circulating strains, whereas the South African strain B.1.351 (Beta variant) and the Brazilian variant P.1 emerged in the Amazon region (Gamma variant) presented multiple mutations in the spike protein, such as the E484K mutation in the RBD, that was demonstrated to allow evasion from neutralizing antibodies [44]. In fact, vaccine breakthrough infections with SARS-CoV-2 variants were reported also in individuals with high titers of neutralizing antibodies [45]. The most recently emerged variants, such as the B.1.617.2 (the so-called Delta variant) and the B.1.1.529 (Omicron), showed both increased transmissibility and immune escape. These characteristics are extremely enhanced with Omicron; this variant presents more than 30 mutations in the spike protein, extreme contagiousness, and a risk of hospitalization reduced to one-third when compared to Delta [46].

### 3.4. Do We Need an Adapted Vaccine?

The impact of variants is likely to vary by product, and imperfect vaccines may probably accelerate immune escape. In general, vaccination should induce the highest neutralization titer possible to maximize protection against antigenically drifted strains [47]. Most vaccines based on the Wuhan sequence showed reduced neutralizing activity against the Beta variant, which was considered the most concerning emerging variant in the first phase of the vaccination campaigns [48]. However, conflicting results were obtained by population-based studies that used different products and different end-points to estimate vaccine effectiveness. For example, a study conducted in Qatar showed a reduced protection of RNA vaccines against infection due to the Beta variant, but protection higher than 90% against severe disease [49], whereas a study conducted in South Africa found no protection induced by a simian adenovirus vectored vaccine against mild to moderate COVID-19 due to the Beta variant [50].

With regard to Delta, UK data suggested two doses of COVID-19 vaccines to be rather effective [51], while the effectiveness was significantly lower after one dose of the vaccine against this variant when compared with the Alpha variant [52]. For this reason, the completion of the vaccination cycle was accelerated to reduce the time interval between the two doses and to provide better protection at the individual and community level. In this regard, a recent meta-analysis found pooled vaccine effectiveness against the Delta variant to be 63.1% against asymptomatic infection, 75.7% against symptomatic infection, and 91% against hospitalization; overall, compared with the Alpha variant, the vaccine protection was slightly reduced against mild disease but fully maintained against severe COVID-19 [53]. Furthermore, after a complete vaccination cycle, neutralizing antibody titers appeared to be 3- to 5-fold lower against the Delta variant compared with the Alpha variant [54].

The recently emerged Omicron variant appears now to have an even higher immune escape potential compared to Delta, as suggested by the decreased neutralizing activity of sera from previously infected or vaccinated individuals [55,56,57]. Vaccine effectiveness is also substantially affected [58], and primary immunization with two vaccine doses shows limited or no protection against symptomatic disease [46,59]. Thus, in order to maintain high levels of protection in the community, the close monitoring of emerging variants and waning immunity is key in order to rapidly move to update vaccines for variants of concern with breakthrough potential, ensuring access to the most up-to-date boosters. Moreover, since higher rates of immunity due to vaccines may create selective pressure which might favor variants that are able to infect people who have been immunized, vaccinating quickly and thoroughly is particularly important [20].

### 3.5. Planning for Booster Doses: Why, Who, When

Apart from immunocompromised patients, who need a third dose to complete the primary cycle, a booster dose is requested to reinforce the protection from infection and disease in individuals immunized a few months before. In fact, the rates of SARS-CoV-2 infection and severe COVID-19 appear to increase as a function of time since vaccination, suggesting a strong effect of waning immunity [60]. In particular, after the Delta variant became predominant, the protection against hospitalization usually remained high over 6 months since primary vaccination, while the protection against infection tended to decrease to less than 50%; however, with the Delta variant, the reduction in vaccine effectiveness over time was probably primarily due to waning immunity rather than to immune escape [61,62].

Studies conducted in the pre-Omicron era, however, showed that a booster (third) dose may restore vaccine effectiveness, reducing rates of confirmed COVID-19 and severe disease also in older individuals already vaccinated with two doses [63,64]. Compared with the second dose, the vaccine effectiveness after the third dose was estimated to be 93% against hospitalization, 92% for severe disease, and 81% for death [65].

Due to the higher immune evasion potential of the Omicron variant, the effect of waning immunity is likely to be amplified, though the protection against hospitalization and severe disease may be conserved. A booster dose of the mRNA vaccine appears to increase neutralizing activity against the Omicron variant [66,67], increasing protection against symptomatic disease and/or hospitalization; however, protection against symptomatic Omicron infection is lower than with Delta, and the effect might decline over time [46,59,68].

Adapted vaccines will become available probably at the end of the summer of 2022. A further booster dose (the so-called fourth dose) may restore antibody levels [69], increasing effectiveness against all COVID-19-related outcomes [70]; however, the protection against infection appears to be short-lived [71].

## 4. Ensuring Access to Vaccines for Global Epidemic Control

Since a global mass immunization program is requested to put the epidemic under control, equity in vaccine allocation and distribution across countries, with particular regard to poor-resource countries, needs to be ensured [72]. However, production constraints and hoarding (i.e., governments may be allowed to force manufacturers to sell domestically) could limit vaccine supplies [73]. The availability of vaccines that require the usual cold chain (2–8 °C) vs. those requiring the extreme cold chain (−20 or −75 °C) may also influence the feasibility and the speed of mass vaccination campaigns. Thus, governments and private funders should invest, taking the risk to give manufacturers money to scale up their production capacity in advance, even if these facilities are never used. Improving access to vaccines and supporting the development of vaccines that can be stored at room temperature for a long time may not only reduce the burden of disease in poor-resource countries but may also advantage wealthier countries. In fact, although there is evidence that a persistent SARS-CoV-2 infection in immunocompromised hosts plays a major role in the emergence of more transmissible/immune escape variants [74], rapid virus circulation in high-density areas might also favor the emergence of such variants, which may be thus introduced in industrialized countries, jeopardizing the success of vaccination campaigns [75].

## 5. Conclusions

Effective vaccines against SARS-CoV-2 have been developed with unprecedented speed to respond to the pandemic threat [76]. However, it is clear that the effects of vaccines on the population cannot be immediate [75]. Vaccination plans have taken into account that only a limited number of doses are available in the first phase of the campaign, requiring the prioritization of specific target populations, with the aim to reduce morbidity and mortality in the short term, and to preserve the healthcare system. Immediately after, adaptive strategies that take into account improved knowledge of the drivers of virus circulation, the impacts of different vaccines on specific population groups, the evolution of the epidemic dynamics, and the need and timing of booster doses of the vaccine possibly adapted to immune escape variants should be designed and implemented in order to accelerate epidemic mitigation and/or containment.

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
