# Peer review of "COVID-19 Vaccination Strategies and Their Adaptation to the Emergence of SARS-CoV-2 Variants"

_vaccines, 2022, doi:10.3390/vaccines10060905_

Round 1

Reviewer 1 Report

The manuscript “COVID-19 vaccination…” compares vaccination strategies during the COVID-19 pandemic in different countries and outcomes of these different approaches. The main topic of the manuscript, as its title indicate, is adaptation of vaccination strategies to deal with emerging variants of SARS-CoV-2 with increased ability to avoid immune response after vaccination. The strategies described are analyzed using statistical data on preventing transmission and severe outcomes after infection occurs with proper references to data. This description can be improved in several ways:

  1. Rather than using initial designations for the vaccines (lines 106, 110), it would be better to use standardized names provided at https://covid19.trackvaccines.org/agency/who/.
  2. Noting that “55% the Europeans said they were concerned about potential side effects of a vaccine”, lines 126-7, authors do not mention that there are real, not potential, safety issues for the vaccines, which led governments to issue warnings and precautions on specific vaccines and which affected decision of being vaccinated by specific age groups. This information is readily available from public health web sites and it has to be discussed in the text among other considerations.
  3. Immune protection due to a vaccination or disease is highly debated issue which is updated on a regular basis. Providing details and percentages published in papers on studies dealing with “population” of a specific country (e.g. Israel, line 167 or US, line 269) does not provide wide coverage as many studies are published on this subject. It would be better to summarize these data in table format which will allow to present a variety of detailed data for age groups, types of vaccine, virus variants etc.
  4. Some assumptions look far-fetched without proper justification. For example, it is stated that “rapid virus circulation in high density areas is likely to favor emergence of immune escape variants” while there are published cases of people who had long-COVID that resulted in mutations without any virus circulation so this option has to be considered as well
  5. Figure 1, which pretends to provide information on the effect of time, looks completely artificial and this fact is recognized by the authors who mention in the figure legend that “Long-term effect of the booster dose cannot be predicted”, lines 288-9. I would suggest removing it.

Author Response

1. Rather than using initial designations for the vaccines (lines 106, 110), it would be better to use standardized names provided at https://covid19.trackvaccines.org/agency/who/.

Reply:  Thanks to the Reviewer. Standardized named are now provided in the text.

2 .Noting that “55% the Europeans said they were concerned about potential side effects of a vaccine”, lines 126-7, authors do not mention that there are real, not potential, safety issues for the vaccines, which led governments to issue warnings and precautions on specific vaccines and which affected decision of being vaccinated by specific age groups. This information is readily available from public health web sites and it has to be discussed in the text among other considerations.

Reply: A sentence raising the comment done by the Reviewr has been added in the 2nd paragraph of the section 2.1 of the paper.

3. Immune protection due to a vaccination or disease is highly debated issue which is updated on a regular basis. Providing details and percentages published in papers on studies dealing with “population” of a specific country (e.g. Israel, line 167 or US, line 269) does not provide wide coverage as many studies are published on this subject. It would be better to summarize these data in table format which will allow to present a variety of detailed data for age groups, types of vaccine, virus variants etc.

Reply:  We got the Reviewer point. However, we think that available data do not permit the construction of a simple data without performing a systematic review (and possibly a metaanalysis). Thus, we tried to avoid more presentation of the data, preferring to provide concepts. Furthermore, we added some references to provide a wider perspective of available knowledge (see rephrasing and changes in section 3.5).

4. Some assumptions look far-fetched without proper justification. For example, it is stated that “rapid virus circulation in high density areas is likely to favor emergence of immune escape variants” while there are published cases of people who had long-COVID that resulted in mutations without any virus circulation so this option has to be considered as well

Reply: The sentence has been modified in accordance with the reviewer suggestion, and a relevant reference has been added.

5. Figure 1, which pretends to provide information on the effect of time, looks completely artificial and this fact is recognized by the authors who mention in the figure legend that “Long-term effect of the booster dose cannot be predicted”, lines 288-9. I would suggest removing it.

Reply:  The Reviwer is right. Thus we removed Figure 1.

Reviewer 2 Report

I have no objection in general. I would like you to include the following points. And, if you have original data, it is better to add in the manuscript.

(1) The need for education on vaccines in general that will help to eliminate prejudice and misconceptions about vaccines for the future.

(2) Improvement of lifestyle to prevent the spread of infectious diseases.

(3) Development of vaccines that can be stored at room temperature for a long period of time so that vaccines can be administered in countries with poor medical facilities.

(4) Possibility of mass immunization through vaccination.

(5) From the standpoint of immunology, even if neutralizing antibody titers drop, memory B cells remain, so immune memory should persist. Since CTLs are also involved in immunity, it is difficult to talk about immunity only in terms of neutralizing antibody titer.

Author Response

(1) The need for education on vaccines in general that will help to eliminate prejudice and misconceptions about vaccines for the future.

Reply: A sentence has been added in the text (2nd paragraph of the section 2.1 of the paper).

(2) Improvement of lifestyle to prevent the spread of infectious diseases.

Reply: We agree with the reviewer on the importance of these points that, however, are implicitly considered throughout the paper.   

(3) Development of vaccines that can be stored at room temperature for a long period of time so that vaccines can be administered in countries with poor medical facilities.

Reply: The point raised by this reviewer is now considered in section 4 of the paper

(4) Possibility of mass immunization through vaccination.

Reply: We agree with the reviewer on the importance of these points that, however, are implicitly considered throughout the paper.   

(5) From the standpoint of immunology, even if neutralizing antibody titers drop, memory B cells remain, so immune memory should persist. Since CTLs are also involved in immunity, it is difficult to talk about immunity only in terms of neutralizing antibody titer.

Reply: B memory cell persistence is nor cited in the text and a specific reference has been included